# A Comprehensive Study on a Deep-Learning-Based Electrocardiography Analysis for Estimating the Apnea-Hypopnea Index

**DOI:** 10.3390/diagnostics14111134

**Published:** 2024-05-29

**Authors:** Seola Kim, Hyun-Soo Choi, Dohyun Kim, Minkyu Kim, Seo-Young Lee, Jung-Kyeom Kim, Yoon Kim, Woo Hyun Lee

**Affiliations:** 1Ziovision Inc., Chuncheon 24341, Republic of Korea; seola.kim@outlook.com (S.K.); choi.hyunsoo@seoultech.ac.kr (H.-S.C.); dohyeon.kim@ziovision.co.kr (D.K.); minkyu.kim@ziovision.co.kr (M.K.); 2Department of Computer Science and Engineering, Seoul National University of Science and Technology, Seoul 01811, Republic of Korea; 3Department of Computer and Communications Engineering, Kangwon National University, Chuncheon 24341, Republic of Korea; 4Department of Neurology, Kangwon National University Hospital, College of Medicine, Kangwon National University, Chuncheon 24289, Republic of Korea; leeseoyoung@kangwon.ac.kr; 5Interdisciplinary Graduate Program in Medical Bigdata Convergence, Kangwon National University, Chuncheon 24341, Republic of Korea; modest14@kangwon.ac.kr; 6Department of Computer Science and Engineering, Kangwon National University, Chuncheon 24341, Republic of Korea; yooni@kangwon.ac.kr; 7Department of Otolaryngology, Kangwon National University Hospital, College of Medicine, Kangwon National University, Chuncheon 24289, Republic of Korea

**Keywords:** sleep apnea, hypopnea, sleep-related breathing disorder, apnea–hypopnea index, electrocardiography, deep learning, convolutional neural network, gated recurrent unit, sleep scoring systems

## Abstract

This study introduces a deep-learning-based automatic sleep scoring system to detect sleep apnea using a single-lead electrocardiography (ECG) signal, focusing on accurately estimating the apnea–hypopnea index (AHI). Unlike other research, this work emphasizes AHI estimation, crucial for the diagnosis and severity evaluation of sleep apnea. The suggested model, trained on 1465 ECG recordings, combines the deep-shallow fusion network for sleep apnea detection network (DSF-SANet) and gated recurrent units (GRUs) to analyze ECG signals at 1-min intervals, capturing sleep-related respiratory disturbances. Achieving a 0.87 correlation coefficient with actual AHI values, an accuracy of 0.82, an F1 score of 0.71, and an area under the receiver operating characteristic curve of 0.88 for per-segment classification, our model was effective in identifying sleep-breathing events and estimating the AHI, offering a promising tool for medical professionals.

## 1. Introduction

Obstructive sleep apnea (OSA) is a global health concern characterized by repetitive upper airway closure leading to intermittent hypoxia during sleep. Approximately 200 million people worldwide are estimated to have OSA [1]. Despite its prevalence, the majority of patients remain undiagnosed [2] due to the inconvenience and complexity of polysomnography (PSG), the current gold-standard test for OSA diagnoses [3]. PSG is known to be a cumbersome process requiring extensive equipment, ample space, and a time-consuming manual scoring process by trained physicians. The manual scoring process has been recognized to be subjective in nature, making inter-rater reliability an important consideration [4,5]. Sleep scoring is necessary not only once for the initial diagnosis but needs to be conducted repeatedly for choosing and evaluating the effect of the treatment of sleep apnea. For example, continuous positive airway pressure (CPAP) is currently accepted as a first-line treatment for OSA but is not always applicable. When CPAP cannot be applied to a patient, alternatives should be considered, such as behavioral treatments, oral appliances, and/or surgery. However, due to the aforementioned limitations of PSG, it is difficult to re-evaluate the patient status to determine the efficacy of OSA treatments. Therefore, there is a pressing need for an automated sleep scoring system capable of evaluating both the existence and severity of sleep-related breathing disorders in patients. The apnea–hypopnea index (AHI), which serves as the standard metric for this assessment, represents the average number of apnea and hypopnea events occurring per hour of sleep. Apnea is defined as a diminution in the peak thermal sensor signal to 90% or below the baseline for at least 10 s. In parallel, hypopnea is characterized by a substantial reduction in airflow of at least 30%, lasting for a minimum duration of 10 s, and is associated with either a reduction in oxygen saturation of more than 3% or an arousal event detectable via electroencephalography. Our proposal is centered around a deep-learning model designed to predict the AHI utilizing more accessible data inputs, such as single-lead electrocardiography (ECG), which can be collected using a portable device.

The main objective of this research was to devise an automated system capable of functioning as an efficient initial screening tool, one that not only precisely classifies each segment, whether it includes sleep events, but also accurately estimates the AHI score. This could potentially facilitate earlier interventions and lead to enhanced outcomes for patients. While numerous investigations have leveraged deep-learning algorithms to differentiate apnea from normal segments using single-lead ECG datasets with sparse annotation [6,7,8], there remains a notable scarcity of research focused on AHI estimation utilizing large and densely annotated datasets. Furthermore, the prevailing methodologies predominantly address sleep apnea, neglecting the significance of hypopnea events in the computation of the AHI. Our study sought to bridge this gap.

We employed a rigorous method for segmenting and labeling the original ECG data to reflect the continuous aspect of sleep scoring. In this way, we could construct an overlapped dataset without risking the distortion of the number of sleep-breathing events in a recording. Our segmenting and labeling strategy is detailed in Section 2. We also proposed a novel deep-learning model that consisted of a feature extractor and a sequence processor. The deep-learning model we proposed synergistically combines the state-of-the-art convolution neural network (CNN) known as the deep-shallow fusion network for sleep apnea detection network (DSF-SANet) [9] and the gated recurrent unit (GRU) [10]. The DSF-SANet is shown to be adept at extracting features and performing segment-wise classification. However, considering the inherent structural limitations of CNNs, it can be less effective in encompassing the complete sequential aspect of a recording-wise ECG analysis when used alone. ECG data, when viewed as time-series data, exhibits temporal dynamics that CNNs may fail to capture. On the other hand, the GRU, a variant of recurrent neural network (RNN), is specifically designed to analyze sequential data. As such, our model was structured so that the one-minute ECG segment is transformed into a feature vector using the CNN, and then the series of the feature vectors is fed into the RNN as a sequence, producing a probability score for each step that includes sleep-breathing events.

Our approach uses the R-R interval (RRI) and R peak amplitude (RPA) as the input data for the model, which are widely considered useful features in machine-learning or deep-learning approaches for sleep apnea detection using an ECG [6,11,12,13]. Both signals reflect the changes in the autonomic nervous system and are known as reliable markers that reflect the effects of sleep-breathing events on the heartbeat. During sleep-breathing events, oxygen desaturation following the cessation of breathing causes sympathetic nervous activation and arousal; subsequently, to prevent hypoxic damage, hyperventilation is followed by an abrupt increase in the tone of the parasympathetic nerve. Variations in the RRI can serve as indicators of changes in the balance between the sympathetic and parasympathetic nervous systems. The RPA, on the other hand, can provide information about the heart’s electrical activity. For instance, deep inhalation can lead to an increase in the heart’s electrical activity and, thus, an increase in the RPA. Exhalation, on the other hand, can lead to a decrease in the RPA.

The main contributions of our research are as follows:
A large-scale dataset with precise annotations: A dataset of over 1000 recordings with precise annotations enhances the robustness and accuracy of our AHI predictions;Accurate AHI predictions for entire sequences: Beyond segment-wise event predictions, our methodology delivers accurate AHI predictions for entire recordings, aligning directly with diagnostic standards;The inclusion of hypopnea as a sleep-breathing event: Our research takes a comprehensive approach, classifying segments into normal and abnormal categories based on sleep-breathing events and integrating both apnea and hypopnea rather than solely differentiating between sleep apnea and normal segments.

## 2. Materials and Methods

### 2.1. Dataset

This retrospective study was reviewed and approved by the Ethics Committee of Kangwon National University Hospital (IRB No. KNUH-A-2022-08-008), and written informed consent was exempted by the Institutional Review Board. The ECG dataset used to train and evaluate our deep-learning model contained 1465 recordings of 1381 patients who were admitted to the Korea National University Hospital (KNUH) sleep center between July 2003 and October 2022. All recordings included a single-lead ECG as part of the PSG tests conducted and scored at the KNUH. All signals were detected from 11 p.m. to 7 a.m. under the supervision of an experienced technician using standard electrodes and sensors recommended in the American Academy of Sleep Medicine manual. All sleep parameters were manually interpreted by a technician and reviewed by a certified physician according to the standard criteria of the 2017 American Academy of Sleep Medicine manual for scoring sleep and related events (v2.4) [14].

Each recording was collected with a sampling rate of 200 Hz. The scoring information for each recording was provided as a text file containing a list of events and their corresponding start time and duration in seconds. The dataset of 1465 recordings was partitioned into training and test sets, including 1220 and 245 recordings, respectively. The training set was further subdivided into training and validation sets according to a 5-fold cross-validation approach. In this process, each fold was constructed independently to ensure that no patients in the training set would appear in the corresponding validation set. The model was trained for each fold and evaluated using test data.

### 2.2. Data Preprocessing

First of all, we downsampled the ECG recordings at 100 Hz to make the following preprocessing steps more efficient. The recordings were then subjected to a bandpass filter to retain the frequency components between 3 and 45 Hz to remove systematic noises such as baseline wander.

We defined the main problem as a per-segment binary classification in which the system classified segments into those with or without sleep-breathing events, including apnea and hypopnea. The predicted AHI score could be calculated afterward per recording, aggregating the per-segment scores produced using the deep-learning model. To construct a dataset accordingly, we segmented each recording into 10-s segments and labeled them based on the scoring information. To ensure that an event segment contained only the event-relevant part, we introduced a rigorous labeling strategy: only 10-s segments that were entirely included in an event period provided in the PSG annotation were labeled as events and were otherwise normal, as shown in Figure 1. This approach was viable because of the availability of the continuous annotation file of each event in the KNUH dataset.

To fully exploit the context information, we extended each 10 s segment to 1-min and 5-min lengths. We first added the preceding and following 25 s part to each 10 s segment 
$x_{t}$
, resulting in a 1 min segment 
$x\prime_{t}$
, as presented in Figure 2. We then created 5 min segments 
$x{\prime \prime }_{t}$
 by pairing each 1 min segment with its two preceding and two succeeding segments. The labels for these 5 min segments corresponded to their central 1 min segment. In this way, we created an overlapped dataset, with a window size of 1 min and a step size of 10 s for the 1 min segments and with a window size of 5 min and a step size of 10 s for the 5 min segments. Despite the overlapped dataset, we could align the actual event counts and the number of event segments simply by aggregating the continuous event segments as a single event. This assumption was justified by the fact that the duration of most sleep-breathing events ranged from 10 to 20 s.

We then separately processed both the 1 and 5 min segments to extract the R-R interval (RRI) and R peak amplitude (RPA) signals, which are typical respiratory signals derived from ECG signals. We used the Christov method [15] for R peak detection for this process. We then interpolated the RRI and RPA signals into 180 data points for the 1 min segments and 900 data points for the 5 min segments. The converted 1 and 5 min segments are denoted as 
$r\prime_{t}$
 and 
$r{\prime \prime }_{t}$
, respectively. We input 
$r\prime_{t}$
 and 
$r{\prime \prime }_{t}$
 into its own dedicated pathway in the model, with each pathway receiving two data channels: RRI and RPA. Figure 3 depicts the original ECG segment along with their corresponding RRI and RPA segments. 

### 2.3. Model Architecture

The overall architecture of the model is illustrated in Figure 4. The model consisted of two main components: a feature extractor and a sequence processor. The feature extractor was modeled after the DSF-SANet, a CNN-based model that demonstrates state-of-the-art performance in sleep apnea detection [9]. On the other hand, the sequence processor was based on a gated neural network, which is an advanced model of a recurrent neural network (RNN). A RNN is widely used for tasks engaging sequential data, such as time-series prediction [16,17,18] or natural language processing [19,20] due to its capability to deal with inter-time dependencies and is thus considered appropriate to leverage the sleep-related context within ECG signals [21].

The feature extractor was based on the CNN. Convolution is a widely employed feature extraction method [22,23] in which a small matrix or kernel is slid across the input to produce aggregated representations that capture useful local information for the intended task. Our feature extractor was composed of three components: the deep feature module 
$F_{D}$
 operating on the 1 min segment 
$r\prime_{t}$
, the shallow feature module 
$F_{S}$
 processing the 5 min segment 
$r{\prime \prime }_{t}$
, and the fusion module 
$F_{F}$
, which performed feature fusion on the integrated deep and shallow features. When 
$r\prime_{t}$
 and 
$r{\prime \prime }_{t}$
 were input to the model, 
$r\prime_{t}$
 progressed to 
$F_{D}$
 and 
$r{\prime \prime }_{t}$
 and progressed through 
$F_{S}$
 to produce deep and shallow features. The two types of features were concatenated and subjected to the fusion module 
$F_{F}$
, which used the attention mechanism [24] to assign importance to each channel of the fused feature and recalculated the feature matrix based on this importance. Finally, the fully connected layer 
$f_{c}$
, transformed the feature matrix into a one-dimensional feature vector with a length of 128. The final resulting feature vector is denoted as 
$h_{t}$
. This is represented using Equation (1) as the following:
(1)
$$h_{t}=F\left(r\prime_{t},\;r{\prime \prime }_{t}\right)=fc(F_{F}\left(F_{D}\left(r\prime_{t}\right)|F_{S}\left(r{\prime \prime }_{t}\right)\right).$$


When a series of inputs of length *N*, 
$X=\{x_{t}|1\leq t\leq N\}$
 passes through the feature extractor 
$F\left(x\right)$
, a series of feature vectors 
$H=\{h_{t}|1\leq t\leq N\}$
 is obtained. This sequence of features *H* is then fed into the sequence processor.

The sequence processor *G* takes the feature sequence *H* fed from the feature extractor and generates a series of outputs 
$O=\{o_{t}|1\leq t\leq N\}$
. Then, a fully connected layer and sigmoid function were successively applied to the output to obtain the apnea scores. Each segment was classified as normal or breathing events based on its score. As a result, the prediction at time *t* is provded as follows, where *σ* indicates the sigmoid function:

(2)
$$y_{t}=\sigma \left(fc\left(G\left(h_{t}\right)\right)\right).$$


### 2.4. Training

The model implementation and experimentation were conducted using Python 3.9.16 and Pytorch 1.12.0. We trained the model for 1000 epochs with early stopping with a warmup of 10 epochs, a batch size of 16 recordings, and an initial learning rate of 10^−4^. The learning rate was scheduled with a factor of 0.1 and patience of 10 epochs. We used the Adam optimizer [25] with betas of 0.9 and 0.999.

The training process followed 5-fold cross-validation; thus, the model was trained five times using different training and validation sets. In each trial, we made sure that no recording of the patients in the validation set appeared in the training set. We obtained five sets of parameters for our model, each of which was then evaluated on the test dataset, which was set aside from the 5-fold training and validation datasets. The mean and standard deviation of each performance metric were calculated.

### 2.5. Evaluation

The evaluation in this study was completed in a two-stage paradigm. First, we evaluated the segment-wise performance of our deep-learning model in terms of accuracy, specificity, sensitivity, F1 score, and area under the receiver operating characteristic curve (AUROC), all of which are widely accepted evaluation metrics for data classification tasks [26]. These metrics evaluate the ability of the model to correctly tell whether each of the segments includes any breathing event or not. The accuracy, specificity, and sensitivity were defined in terms of true positives (TPs), true negatives (TNs), false positives (FPs), and false negatives (FNs) as follows [26]:

(3)
$$\mathrm{Accuracy}=\frac{\mathrm{TP}\;+\;\mathrm{TN}}{\mathrm{TP}+\mathrm{TN}+\mathrm{FP}+\mathrm{FN}},$$


(4)
$$\mathrm{Specificity}=\frac{\mathrm{TN}}{\mathrm{TN}+\mathrm{FP}},$$


(5)
$$\mathrm{Sensitivity}=\frac{\mathrm{TP}}{\mathrm{TP}+\mathrm{FN}}.$$


Here, TP refers to instances where the model accurately identified the presence of a breathing event within a segment. TN indicates scenarios where the model correctly recognized the absence of any event in a segment, aligning with the actual absence of events. On the other hand, FP denotes situations where the model erroneously classified a segment as containing an event, despite the segment being event-free. Lastly, FN describes cases where the model failed to detect an event, even though an event did indeed occur within the segment. The F1 score is the harmonic mean of the sensitivity and specificity, providing a balanced measure of a model’s performance regarding false positives and negatives. The AUROC is a performance indicator based on the trade-off between sensitivity and specificity at various classification thresholds, above which the segments are classified as positive. The AUROC is often considered a vital evaluation metric in classification tasks for imbalanced datasets because it offers a comprehensive and distribution-insensitive measure of a model’s discriminatory power, enabling more informed decisions in model selection.

Secondly, we assessed the performance of our deep-learning model on a per-recording basis, determining whether the model accurately predicted the apnea–hypopnea index (AHI) for each recording. To accomplish this, we calculated the AHI from the metadata and annotations—referred to as the true AHI—as well as the target AHI value, a hypothetical value calculated from the target labels in the dataset of 1-min segments. It is the value of the predicted AHI assuming a perfect model, representing what our model aims to predict. Due to the model’s segment-based approach to event detection, some differences between the true AHI and target AHI were to be expected. The predicted AHI was calculated in the same way as the target AHI, except using the predictions of our model instead of the target labels in the dataset. The target and predicted AHI values were derived by counting the segment-wise number of events (whether based on the target label or predictions), while the total sleep time was inferred from the total number of segments within a recording. In Section 3, we present a comparison between the true AHI, the target AHI, and the predicted AHI.

An additional set of metrics was designed to assess the capacity of the model to accurately estimate the AHI. The Pearson correlation coefficient [27], root mean square error (RMSE), and mean absolute error (MAE) were used to quantify the discrepancy between the estimated and actual values in predictive analytics or modeling [26]. The correlation coefficient gauges the linear relationship between two variables and is defined as the following:

(6)
$$r_{xy}=\frac{Cov\left(x,\;y\right)}{\sigma_{x}\sigma_{y}},$$

where 
$Cov\left(x,y\right)$
 refers to the covariance of variables *x* and *y* and 
$\sigma_{x}$
 and 
$\sigma_{y}$
 signify the standard deviations of *x* and *y*, respectively. The MAE serves as a measure of the average magnitude of errors throughout the entire prediction, where the size of the errors is determined using the absolute difference between the predicted and actual values, as shown in the following equation [26]:

(7)
$$\mathrm{MAE}=\frac{\sum_{i=1}^{n}\left|y_{i}-x_{i}\right|}{n}.$$


The RMSE also gauges the discrepancy between the predicted and actual values, but the difference between each value is initially squared, then summed, and finally, the square of this sum is taken to be the RMSE. The definition is provided in the following equation [26]:

(8)
$$\mathrm{RMSE}=\sqrt{\frac{\sum_{i=1}^{n}{\left(y_{i}-x_{i}\right)}^{2}}{n}}.$$


These metrics provide insight into how closely our predictions aligned with both the actual and target values on average, with smaller values indicating better alignment.

## 3. Results

### 3.1. Per-Segment Performance

The per-segment performance of the proposed model is detailed in Table 1. Each metric was calculated based on the model’s predictions at each time step of 10 s. The model exhibited robust discriminative ability, denoted by the AUROC values of around 0.88 across both the validation and test sets. It was also observed that there was very little difference between the metrics in the validation and test sets. Upon excluding the hypopnea segments, the performance metrics on a per-segment basis, as detailed in Table 2, demonstrated notable enhancement across all metrics. This improvement suggests the inherent challenges associated with accurately detecting hypopnea events, thereby highlighting the model’s increased efficacy in scenarios devoid of such complexities.

### 3.2. Per-Recording Performance

Figure 5 depicts the scatter plots of the true, target, and predicted AHI values. Here, the “true-predicted AHI” represents the model’s predictions compared to the actual AHI values calculated from the continuous recording data, whereas the “target-predicted AHI” refers to the comparison between the ideal and actual outcomes derived from the discrete segment-wise analysis. The model showed a correlation coefficient of 0.89 for the target-predicted AHI and 0.87 for the true-predicted AHI. The correlation coefficient between the true AHI and target AHI was measured to be 0.98, as can be seen in Table 3. To further evaluate our model’s ability to predict the AHI on the test set, the mean absolute error (MAE) and the root mean square error (RMSE) were also calculated between the true, target, and predicted AHI values. Detailed values of these error metrics, along with the correlation coefficients, are enumerated in Table 3. The MAE showed a difference of 1.57 and 2.10 compared to the ideal case for the target-predicted and true-predicted values, respectively. For the RMSE, the difference was 2.55 and 3.70 for each case.

Example cases with a duration of 150 s are presented in Figure 6. Figure 6a,b outlines scenarios where the events were accurately detected. On the other hand, Figure 6c shows an instance where the model falsely detected the events before and after the actual event segments (overestimation). Figure 6d describes a case where the model only partially detected the events (underestimation). Here, the raw data that were downsampled to 100 Hz are depicted, and the sleep-breathing event detection was made every ten seconds, which was the step size of our overlapped dataset of 1-min and 5-min segments. Although the results are not perfect, the model still holds value as a preliminary screening method because it can provide insights into whether the patient needs to undergo more comprehensive tests or if certain therapeutic interventions are effective.

## 4. Discussion

In our research, we pioneered an automated screening system that adeptly classifies sleep segments and precisely calculates the AHI, thereby facilitating prompt interventions that could significantly ameliorate patient outcomes. The novelty of our approach lies in its meticulous incorporation of both apnea and hypopnea events, leveraging a densely annotated and large dataset, in contrast to previous studies focused on apnea identification with sparsely annotated data. Our methodology’s strength is its ability to deliver robust AHI estimations across entire recordings, aligning seamlessly with established diagnostic criteria. Our deep-learning model, trained on a densely annotated large dataset, not only demonstrated the ability to correctly predict the AHI but also showed excellent generalizability, ensuring its performance across real-world diagnostic settings.

We introduced a deep-learning model that combined a CNN for feature extraction and a GRU for sequence processing. The model was trained on the KNUH dataset and achieved a correlation coefficient of 0.87 between the actual and predicted AHI. This high correlation indicates its ability to identify sleep-related breathing events at the recording level accurately and to distinguish between normal segments and those with sleep-breathing events on the segment level, as evidenced by an AUROC of 0.88. The negligible discrepancies in the performance metrics between the validation and test datasets indicate the model’s strong generalization ability, suggesting that the model learned the underlying patterns and features from the training data.

The KNUH dataset used in this study included ECG data from 1465 patients along with annotations, which was significantly larger than the Apnea-ECG database v1.0.0 [28], a currently available public dataset in the domain of sleep apnea and ECG, which has 70 recordings, with only 35 annotated. Utilizing a sufficient amount of data is crucial to avoid overfitting and ensure the model’s robustness on unseen data. By leveraging a large dataset, we maximized the model’s representational capacity, achieving similar performance on the test set as on the validation set. Moreover, the continuous annotation format of the KNUH dataset, marking the patient status every second as opposed to the more common per-minute annotation, facilitated the development of a dense labeling and segmentation approach. This methodology resulted in a near-perfect alignment between the actual and segmented dataset-derived AHI, as shown by a correlation coefficient of 0.98.

Our method also generated a significantly augmented dataset through the creation of overlapping segments without any artificial transformation that could compromise the integrity of the natural ECG signal. This is considered to have helped overcome the shortcomings of single-lead ECG datasets compared to 12-lead ECG datasets and led the model to learn generalizable features. Instead of raw ECG data, we used the RRI and RPA as inputs, reducing the data size and mitigating noise, which helped the model focus on the meaningful features related to sleep-breathing events. This approach made the task computationally easier for the model and improved its robustness on unseen data.

However, there were limitations to our methodology. Our labeling and segmentation strategy could have led to counting a single event multiple times in adjacent segments if the event lasted longer than 20 s. We addressed this by reducing multiple successive event counts to a single count, which resulted in a slight underestimation of the event counts. Additionally, our model performed better without hypopnea segments, indicating that detecting hypopnea is more challenging than detecting apnea for our model. Future research could focus on improving hypopnea detection or developing more advanced post-processing methods. It would also be beneficial to explore whether better results can be achieved by using raw ECG signals instead of converting them into R-peak-based signals. Furthermore, to meet the demand for more immediate decisions, further validation of our methodology for shorter intervals than a minute is necessary.

## 5. Conclusions

In conclusion, our study presents a novel method for sleep apnea detection using an ECG that integrates the RRI and RPA using a combination of CNNs and GRUs. This method effectively detects the intervals where the sleep-breathing events occur on the segment and recording level, demonstrating the potential of our model’s application in assisting medical professionals and enhancing patient care. One possible application is as a pre-monitoring tool, assisting physicians by providing them with preliminary insights into patients’ conditions before they undergo more extensive tests such as PSG. There is also potential for the development of a home-based sleep monitoring system that incorporates wearable devices capable of ECG recording. Such a system could serve as a preliminary screening tool for sleep-related breathing disorders or as a means to assess the effectiveness of therapeutic interventions.

## Figures and Tables

**Figure 1 diagnostics-14-01134-f001:**
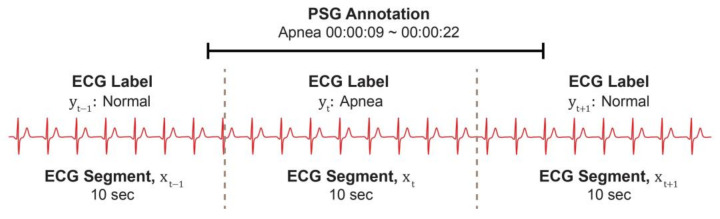
Creating labels for 10 s segments from the PSG annotation. PSG: polysomnography; ECG: electrocardiography.

**Figure 2 diagnostics-14-01134-f002:**
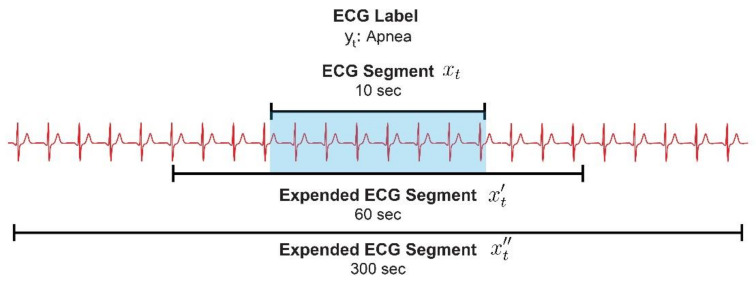
The 10 s ECG segment 
$x_{t}$
, 1 min segment 
$x\prime_{t}$
, and 5 min segment 
$x{\prime \prime }_{t}$
. ECG: electrocardiography.

**Figure 3 diagnostics-14-01134-f003:**
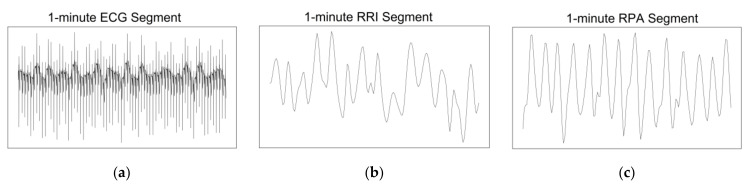
Visualization of 1 min segments of the (**a**) original ECG, (**b**) RRI, and (**c**) RPA. RRI: R-R interval; ECG: electrocardiography; RPA: R peak amplitude.

**Figure 4 diagnostics-14-01134-f004:**
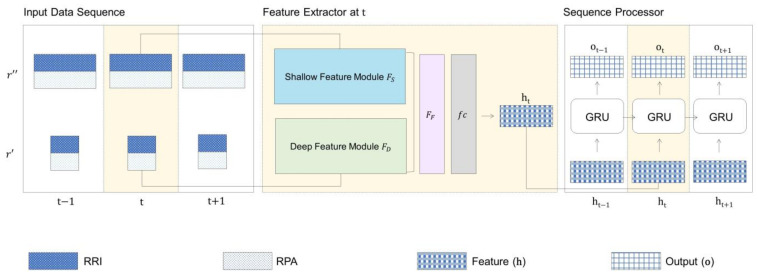
The structure of our model. The input contains two sequences: a sequence of 1 min segments 
$r^{\prime }$
 and 5 min segments 
$r^{\prime \prime }$
. These two sequences are separately processed in the feature extractor to produce a feature vector at the time point *t*. The sequence of the features ***h*** is then fed into the sequence processor, yielding an output sequence ***O***, which results in an apnea score at each time point after passing through a fully connected layer and a sigmoid function. RRI: R-R interval; RPA: R peak amplitude; GRU: gated recurrent unit.

**Figure 5 diagnostics-14-01134-f005:**
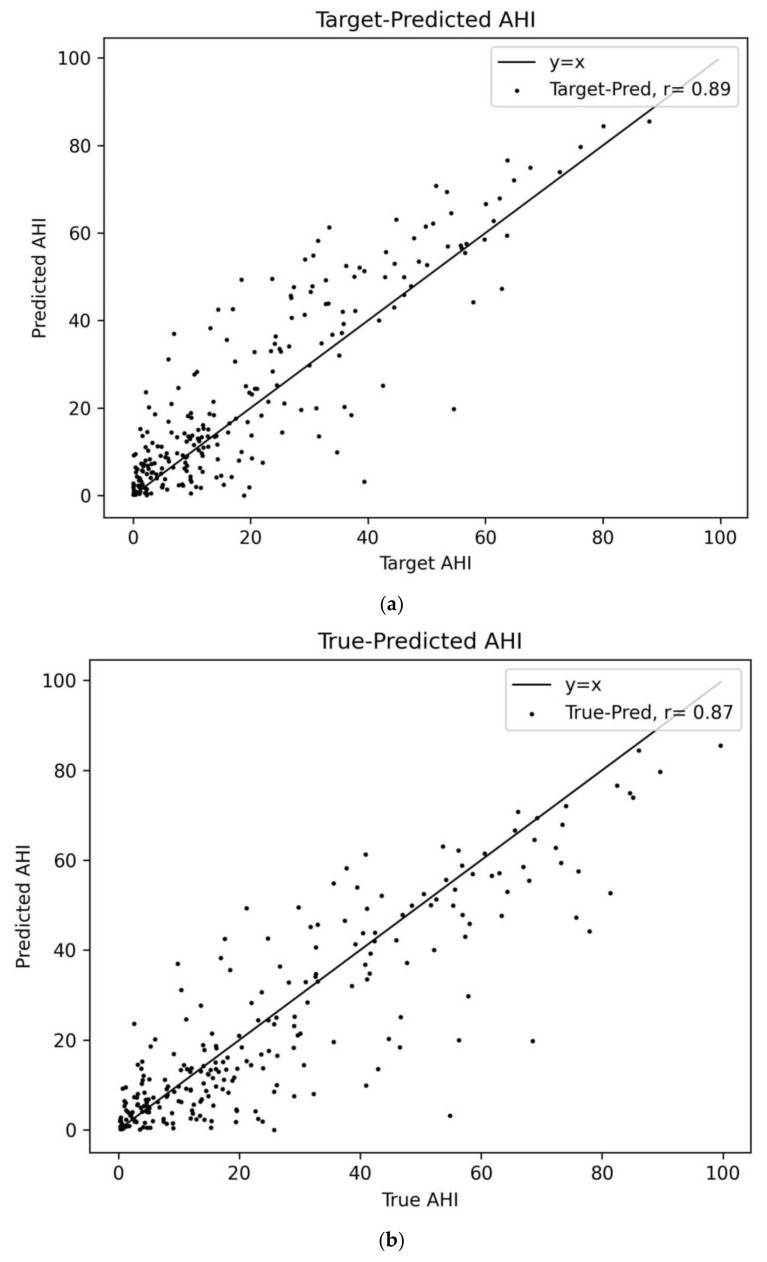
(**a**) The relationship between the target and the predicted AHI and (**b**) the relationship between the true and the predicted AHI. AHI: apnea–hypopnea Index.

**Figure 6 diagnostics-14-01134-f006:**
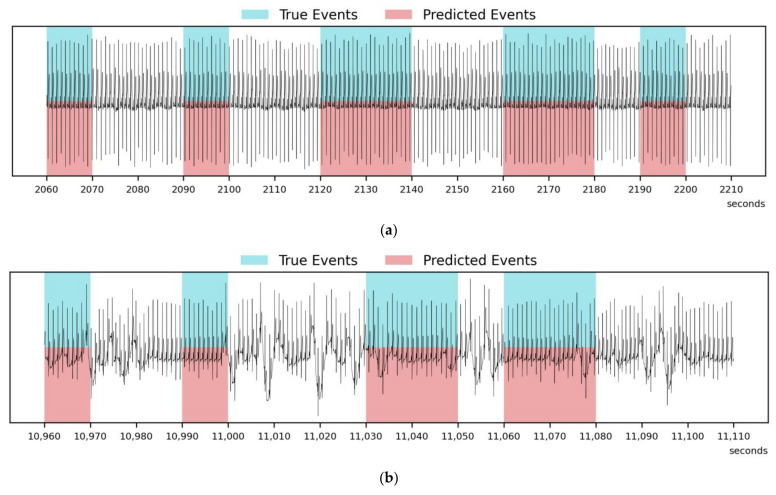
A comparative illustration of 150 s sequences demonstrating (**a**,**b**) accurate event detection, (**c**) overestimated event detection, and (**d**) underestimated event detection.

**Table 1 diagnostics-14-01134-t001:** The per-segment performance of our model. AUROC: area under the receiver operating curve.

Metric	Validation Set	Test Set
Accuracy	0.815 ± 0.009	0.820 ± 0.004
Sensitivity	0.766 ± 0.115	0.774 ± 0.009
Specificity	0.821 ± 0.010	0.827 ± 0.006
F1 Score	0.695 ± 0.016	0.709 ± 0.003
AUROC	0.876 ± 0.008	0.883 ± 0.002

**Table 2 diagnostics-14-01134-t002:** The per-segment performance of our model, considering only the normal and apnea segments. AUROC: area under the receiver operating curve.

Metric	Validation Set	Test Set
Accuracy	0.890 ± 0.007	0.898 ± 0.004
Sensitivity	0.870 ± 0.017	0.876 ± 0.008
Specificity	0.891 ± 0.008	0.899 ± 0.004
F1 Score	0.720 ± 0.025	0.736 ± 0.005
AUROC	0.948 ± 0.007	0.953 ± 0.003

**Table 3 diagnostics-14-01134-t003:** The correlation coefficient, MAE, and RMSE between the actual, target, and predicted AHI values. AHI: apnea–hypopnea index; MAE: mean absolute error; RMSE: root mean square error.

	True Target	Target Predicted	True Predicted
Correlation Coefficient	0.98	0.89	0.87
MAE	5.85	7.42 (+1.57) *	7.95 (+2.10) *
RMSE	7.89	10.44 (+2.55) *	11.59 (+3.70) *

* The parenthesized values represent the difference compared to the true target (ideal) value.

## Data Availability

The data presented in this study are available upon request from the corresponding author due to legal restrictions on privacy and ethical reasons.

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
