# Peer review of "A Comprehensive Study on a Deep-Learning-Based Electrocardiography Analysis for Estimating the Apnea-Hypopnea Index"

_diagnostics, 2024, doi:10.3390/diagnostics14111134_

Round 1

Reviewer 1 Report

Comments and Suggestions for Authors

This study focuses on estimating the apnea-hypopnea index. Herein, a deep learning-based system is proposed to detect sleep apnea using a single-lead electrocardiography signal. In the study, the model trained using 1,465 ECG recordings is analyzed at 1-minute intervals. From the results, it can be seen that the AUC value achieved by the model developed to predict the apnea-hypopnea index is 88%. In light of all this information, I suggest the following corrections for the acceptance of the study:

1- Please check the full paper for grammar.

2- In the Introduction, please provide the contributions of this study item by item.

3- Please also mention the models used in the abstract.

4- Reference is required for some equations. Please provide these.

5- Please provide appropriate references for the evaluation section.

6- Please present the accuracy and f1 score values obtained in the study in the abstract.

7- Please provide the first letters of the subheadings in uppercase.

8- Confusion matrices can be presented in responses to reviewers. Please only provide these in responses to reviewers.

9- The discussion section needs improvement. In this section, it is expected that the studies in the literature and the proposed method will be discussed in an argumentative manner. Please provide these.

10- Please provide a conclusion section in the study.

11- Please mention the limitations and future extensions of the study.

The authors should carefully proofread this manuscript before it is to be resubmitted.

Comments on the Quality of English Language

-

Reviewer 2 Report

Comments and Suggestions for Authors

I have the following comments and recommendations to the paper:

• Abstract: ... ECG signals at 1-minute intervals: Many portable equipments have standard measurement period 30 sec. (I personally – as a cardiac – have a such beurer one.) Are your method reliably able to work with this truncated interval?

• Line 113: Really, do you have records from 2003 (more than twenty years old)?

• Line 120: sampling rate at 200 Hz: This sampling rate is sufficient for “normal” patients. However, in the case of fibrillation, the bpm could be very high, even more than 200 (and more than 300 in extreme cases). Moreover, the shape of the ECG pulses can differ considerably in these cases (especially for ventricular fibrillations). My fundamental question is whether your method is applicable for the patients with atrial fibrillation, or even with ventricular fibrillation. (And the sampling rate probably should be greater in these cases.)

• Line 199: Replace “sequence processor” with “The sequence processor”.

• Line 216: “Adam optimizer” … please insert a citation where it can be found.

• Line 259: “Pearson correlation coefficient” … the same.

• Line 265: Missing space before “signifies”.

• Subsections 3.1 and 3.2: The section names should have an initial capital letter.

• Figure 5: The two ((a) and (b)) figures are too small. Please use the full width of the page to increase the size of the figures.

• Figure 6: Although there can be overestimated or underestimated detections, the method still seems to be useful. I also worked in the bioelectronics area, and I know that the degree of uncertainty is always quite high. It should be emphasized that a partially successful detection is still useful for patient treatments.

Generally, I think the paper can be accepted after a minor revision. 

Comments on the Quality of English Language

The English language is quite OK.
